# Direct single molecule measurement of TCR triggering by agonist pMHC in living primary T cells

Geoff P O'Donoghue[1,2,†], Rafal M Pielak[1,2,†], Alexander A Smoligovets[1,2,3], Jenny J Lin[1,2], Jay T Groves[1,2*]

[1]Department of Chemistry, Howard Hughes Medical Institute, University of California, Berkeley, Berkeley, United States; [2]Physical Biosciences Division, Lawrence Berkeley National Laboratory, University of California, Berkeley, Berkeley, United States; [3]Department of Molecular and Cell Biology, University of California, Berkeley, Berkeley, United States

**Abstract** T cells discriminate between self and foreign antigenic peptides, displayed on antigen presenting cell surfaces, via the TCR. While the molecular interactions between TCR and its ligands are well characterized in vitro, quantitative measurements of these interactions in living cells are required to accurately resolve the physical mechanisms of TCR signaling. We report direct single molecule measurements of TCR triggering by agonist pMHC in hybrid junctions between live primary T cells and supported lipid membranes. Every pMHC:TCR complex over the entire cell is tracked while simultaneously monitoring the local membrane recruitment of ZAP70, as a readout of TCR triggering. Mean dwell times for pMHC:TCR molecular binding of 5 and 54 s were measured for two different pMHC:TCR systems. Single molecule measurements of the pMHC:TCR:ZAP70 complex indicate that TCR triggering is stoichiometric with agonist pMHC in a 1:1 ratio. Thus any signal amplification must occur downstream of TCR triggering.

*For correspondence:
JTGroves@lbl.gov

[†]These authors contributed equally to this work

Competing interests: The authors declare that no competing interests exist.

## Introduction

An essential aspect of adaptive immunity is the ability of T cells to discriminate between structurally similar agonist and non-stimulatory self peptides bound to major histocompatibility complex (pMHC) molecules presented on the surface of antigen-presenting cells (APCs) (*Weiss and Littman, 1994*). Fewer than 10 agonist pMHC molecules can trigger T cells (*Irvine et al., 2002*; *Manz et al., 2011*), and the signaling response occurs on a timescale of seconds (*Huse et al., 2007*). It has been recognized for decades that subtle, peptide-specific differences in binding parameters, especially kinetic rates for the pMHC and T cell receptor (TCR) reaction, seem to be the basis of antigen recognition (*Matsui et al., 1994*; *Tian et al., 2007*). Experimental correlations between molecular pMHC:TCR binding kinetics, typically determined from solution assays of soluble extracellular domains and bulk measurements of T cell activity, have provided the foundation for much of this understanding.

From a physical point of view, the extreme sensitivity, selectivity, and apparent immunity to stochastic noise exhibited by TCR antigen recognition pose challenges to classical notions of ligand-receptor signaling (*van der Merwe and Dushek, 2011*; *Dustin and Groves, 2012*). For example, the maximum affinity a TCR can have for self pMHC ligands (non-triggering) is sharply defined and close to the affinity threshold for foreign pMHC (triggering). Consequently, every TCR is likely to bind a subset of the ever-present self pMHC molecules with affinities differing only slightly from genuine foreign agonist pMHC. If TCR triggering and signaling were simply proportional to TCR occupancy by ligand, then the abundance of self pMHC could easily overwhelm genuine agonist signals, rendering this as an

**eLife digest** The immune system identifies and combats foreign objects, including pathogens, in the body. T cells are key components of the immune system, and each has a unique variant of a signalling complex known as the T cell receptor on its surface. T cells scan the surfaces of other cells in search of antigens, which are peptides (fragments of proteins) that derive from foreign pathogens such as viruses. Successful recognition of a foreign peptide leads to an immune response that, in most cases, ultimately rids the body of the pathogen. Most importantly, however, the immune system must be able to discriminate between peptides that are produced naturally in the body ('self' peptides) and foreign or 'non-self' peptides. This is challenging because self peptides may have similar structures to non-self peptides and are often much more abundant.

Many models have been proposed to explain how T cells are able to detect just a few molecules of foreign peptide. According to some hypotheses the T cell receptors get together in clusters to function cooperatively; alternatively, it has been suggested that rapid binding of a foreign peptide to multiple T cell receptors sequentially can build up a strong signal. However, none of these phenomena have been directly observed.

O'Donoghue et al. now image the interactions between T cell receptors and peptides bound to molecules called major histocompatibility complexes (MHCs), and show that T cell activation can occur when a single foreign peptide binds to a single receptor. These interactions are long-lived and ultimately result in the recruitment of ZAP70, which has an important role in the activation of T cells, to the complex formed by the T cell, the peptide and the MHC molecule. Therefore, any amplification of the activating signal transmitted by non-self peptides occurs following receptor binding, in contrast to previous models.

ineffective discriminatory mechanism. Alternatively, each engagement of pMHC with TCR (serial triggering; *Valitutti et al., 1995*), perhaps with some minimum engagement time (kinetic proof reading; *McKeithan, 1995*; *Rabinowitz et al., 1996*), might define the threshold for TCR triggering. Other proposed mechanisms elaborate further, suggesting (pseudo) heterodimers (*Irvine et al., 2002*; *Krogsgaard et al., 2005*) of self and agonist pMHC molecules or pMHC-independent forms of trans activation (*Cooper and Qian, 2008*) of multiple TCR by a single agonist pMHC may be at work. Proving or disproving any of these various possibilities based on current data is confounded by the vast difference between ensemble biochemical measurements and cell population behavior. Substantial ambiguity with respect to the actual molecular mechanisms responsible for antigen triggering of T cells remains. A physically accurate understanding of this remarkable process will require simultaneous observations of the molecular binding kinetics, stoichiometry, and movement of individual signaling complexes in living T cells.

Here, we characterize the molecular interactions between pMHC and TCR, at the single molecule level, while simultaneously monitoring the local membrane recruitment of cytosolic Zeta-chain-associated protein kinase 70 (ZAP70) in live primary T cells. Every pMHC can be individually resolved and tracked for up to minutes before photobleaching by using a multi-timescale single molecule fluorescence imaging approach. Key to this strategy is the variable control of excitation light dose and exposure time to achieve hardware discrimination of molecular species with different mobilities. The pMHC molecules fall into two unambiguously distinguished classes: one undergoing fast random motion and the second moving slowly along linear trajectories. These slow moving pMHC are the bound pMHC:TCR complexes; they are only observed with agonist peptide and they spatially correlate with both TCR and recruitment of ZAP70. The linear trajectories of the pMHC:TCR complexes match the well-characterized cytoskeleton-driven movement of TCR during the formation of the immunological synapse (*Campi et al., 2005*; *Yokosuka et al., 2005*; *DeMond et al., 2008*; *Yu et al., 2012*). Thus a lone agonist pMHC bound to TCR leads to stable engagement of the resulting complex with the cytoskeleton. Single molecule intensity calibration of the number of ZAP70 recruited to the vicinity of each agonist pMHC indicates that TCR are triggered in a 1:1 stoichiometry with pMHC.

Associations of pMHC with TCR exhibited molecular binding dwell times with mean durations of 53.8 ± 12.2 s and 5.2 ± 0.2 s for AND and 5c.c7 TCRs, respectively. Individual dwell times are roughly exponentially distributed and are in general agreement with bulk solution measurements of pMHC:TCR

kinetic off-rates for both TCRs (*Corse et al., 2010*; *Huppa et al., 2010*; *Newell et al., 2011*). However, dwell times measured from tracking experiments specifically correspond to spatial entrapment of pMHC with a TCR, or cluster of TCRs (*Schamel and Alarcon, 2013*), on the T cell surface. They do not necessarily correspond to individual molecular binding events with a single TCR. Indeed, recent studies (e.g., by FRET) have suggested that pMHC:TCR kinetic off-rates may be accelerated in living cells relative to in vitro measurements, possibly as a result of actively applied forces from the cytoskeleton (*Huang et al., 2010*; *Huppa et al., 2010*; *Zhu et al., 2013*). We explore the possibility that individual dwell times observed by tracking experiments could be composed of rapid unbinding and serial rebinding of pMHC to multiple TCR within a cluster. Results from a stochastic reaction-diffusion analysis, covering a wide range of parameter space, indicate that serial rebinding alone is unlikely to sustain entrapment. If pMHC thoroughly disengages from TCR, it will most likely escape. Structural flexibility within the pMHC:TCR complex (*Adams et al., 2011*; *Hawse et al., 2012*; *Reboul et al., 2012*) could give rise to apparently fast kinetics (e.g., in FRET measurements) without actual unbinding of pMHC from TCR. The tracking observations reported here directly reveal that the functional interaction between agonist pMHC and TCR is long-lived in living cells and that triggered TCR remain localized with the same pMHC.

## Results

### Characterization of agonist pMHC:TCR complexes

We probe agonist pMHC:TCR complex dynamics in hybrid live cell—supported membrane junctions (*Grakoui et al., 1999*) (*Figure 1A*). The supported membrane is functionalized with MHC (IE$^k$) and intercellular adhesion molecule-1 fusion with a blue fluorescent protein (ICAM1-TagBFP), both linked to the membrane via C-terminal poly-His tag binding to Ni-chelating lipids (*Nye and Groves, 2008*). The MHC is loaded with peptide (moth cytochrome c [MCC] agonist or null), which is covalently coupled in a 1:1 stoichiometry (verified by HPLC) to the photostable fluorophores Atto647N or Atto488 using maleimide-thiol chemistry. Upon contact between the T cell and the supported membrane, Leukocyte function-associated antigen 1 (LFA1)-ICAM1 binding leads to rapid cell spreading and formation of an essentially planar interface between the T cell and supported membrane, within which pMHC:TCR interactions occur.

At fast exposure times (17.5 ms) and high excitation powers (0.2 W/cm$^2$), all pMHC molecules are readily resolved (*Figure 1B*, left panel). The pMHC move as individual molecules, as identified by single-step photobleaching, and can be continuously tracked for up to hundreds of frames (*Figure 1C*). In regions without a cell, pMHC exclusively exhibit random diffusive motion. The step-size distribution from these trajectories corresponds to a single lateral diffusion coefficient of 0.44 (SEM = 0.002) μm$^2$/s, which is typical for supported membranes (*Lin et al., 2010*). Within the T cell junction, the step-size distribution becomes bimodal (*Figure 1D*). In addition to a fast component from freely diffusing pMHC, a distinct slow-moving component also appears.

At long exposure times (500 ms) and low excitation powers (0.02 W/cm$^2$), the fast moving pMHC fraction in *Figure 1D* is averaged over several pixels to form a relatively homogenous background. The slow moving molecules remain highly localized and can be unambiguously tracked for longer than a minute, using 1–10 s time-lapses (*Figure 1B*, right panel; *Video 1*). The slow moving pMHC molecules colocalize with TCR (*Figure 2*) and move in linear trajectories toward the geometric center of the live cell-supported membrane junction; these are the pMHC:TCR complexes. When MHC is loaded with a mixture of agonist and null peptides, with different fluorescent labels, only the agonist peptides are observed in the slow-moving complexes (*Figure 3*). The result is identical when the fluorophores are reversed, excluding the possibility that fluorophore effects could be responsible for binding.

The agonist pMHC densities used in these experiments range from 0.2 to 100 molecules/μm$^2$. The lowest densities are near thresholds for triggering Ca$^{2+}$ flux (*Manz et al., 2011*) and below levels where stable TCR microclusters are readily visible (*Figure 2*), whereas the higher densities are well above the densities at which microclusters are observed (*Campi et al., 2005*; *Yokosuka et al., 2005*; *DeMond et al., 2008*; *Yu et al., 2012*). Nonetheless, the observed single molecule pMHC motion at all densities is reminiscent of the well-characterized actomyosin-driven TCR microcluster radial transport in cells that are activated (*Campi et al., 2005*; *Yokosuka et al., 2005*; *Kaizuka et al., 2007*; *DeMond et al., 2008*; *Yu et al., 2012*).

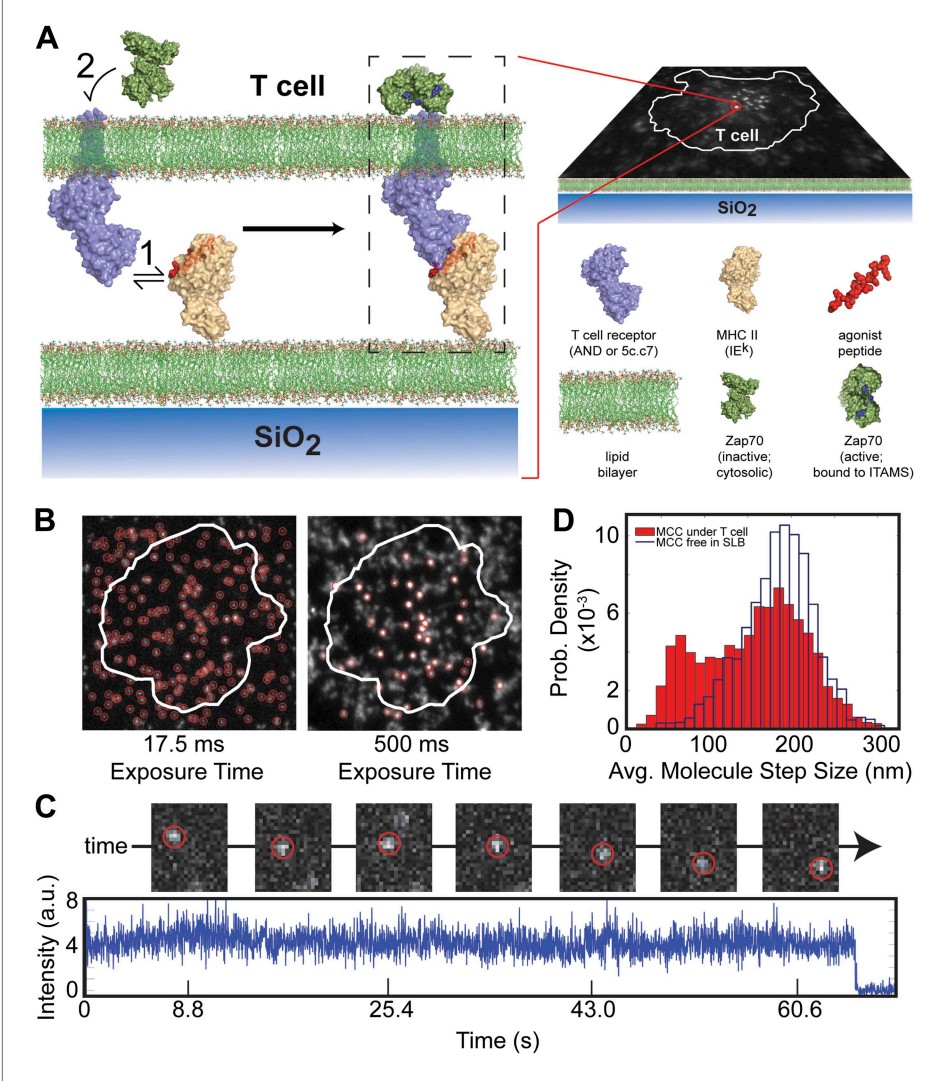

**Figure 1**. Agonist pMHC binding to TCR in T cells is revealed by changes in mobility. (**A**) Schematic of the hybrid live cell–SLB system. Binding of agonist pMHC to TCR (PDB, 3QIU) leads to phosphorylation of ITAMs, on the intracellular domain of the TCR, which is followed by recruitment of the kinase ZAP70 (PDB, 2OQI, 2OZO). We directly observe both pMHC:TCR binding and ZAP70:ITAM recruitment using single molecule fluorescence microscopy (bilayer adapted from **Domanski et al., 2010**). (**B**) At short exposure times (17.5 ms, left) all agonist pMHC molecules are readily resolved. Imaging with a long exposure time (500 ms, right) allows for unambiguous discrimination between the slow, TCR-bound fraction of agonist pMHC and the fast diffusing fraction. This also allows for long (1–10 s interval) time-lapse sequences (**Video 1**). Automated detection of single molecule features (red circles) is discussed in methods. (**C**) Representative intensity trace showing a single agonist pMHC molecule, identified by step photobleaching, bound continuously for ~60 s. (**D**) Step size histogram of single agonist pMHC molecules in a SLB is bimodal under T cells (red) and unimodal before addition of T cells (blue). pMHC molecules in (**B**)–(**D**) were labeled with Atto647N on the MCC peptide.

## TCR triggering monitored by ZAP70 recruitment

Two-color single molecule tracking is used to quantitatively monitor membrane recruitment of cytosolic ZAP70-EGFP (incorporated by retroviral transfection) to the locations of the pMHC:TCR complexes. Using a dual-view system in which chromatic aberrations have been mapped (**Figure 4—figure supplement 1**), spatial colocalization between the two channels to less than 105 nm is achieved. Immediately after cell landing, ZAP70 localizes to and moves together with the pMHC:TCR complexes (**Figure 4A**; **Videos 2 and 3**). For each frame in a tracking sequence, fluorescence intensity

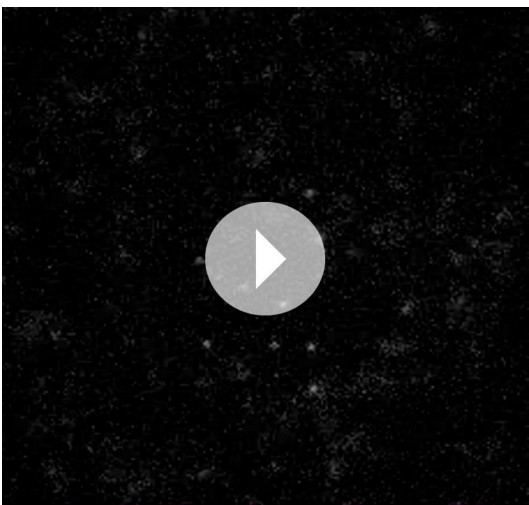

**Video 1**. A 3 s time-lapse video (189 frames) of agonist pMHC interacting with live 5c.c7 T cells (labeled with MCC-Atto488). A long (500 ms) exposure time allows for unambiguous discrimination between TCR-bound and unbound agonist pMHC in the bilayer. The hardware filtering approach utilized here facilitates particle detection in a relatively dense field of fluorophores, allowing for linking particles between frames even when long (1–10s) time lapses are introduced. This technique also has the advantage of using only one probe, compared to other techniques for detecting ligand binding, such as smFRET, which require two different color probes to detect one molecular binding event. These advantages allow for simultaneous two color single molecule tracking and kinetics measurements.

in the ZAP70-EGFP channel is integrated over a 315 nm square region centered on each pMHC:TCR complex (*Figure 4B*). The resulting intensity traces reveal discrete changes in average intensity, which we attribute to the binding of one ZAP70-EGFP to the phosphorylated ITAM domains on the cytoplasmic side of TCR engaged with pMHC (*Iwashima et al., 1994*). Representative traces of colocalized ZAP70-EGFP intensity along with the corresponding intensity trace from the pMHC are illustrated in *Figure 4C*. Stochastic transitions are identified using a change-point algorithm (*Ensign and Pande, 2009*) and results from this analysis are superimposed on the raw intensity traces. Observed molecular binding dwell times of individual ZAP70-EGFP, as resolved by the stochastic transitions, range from 12 to 107 s. The majority of single molecule ZAP70-EGFP traces exhibit intensity fluctuations consistent with background (*Figure 4—figure supplement 2A*), which would not be expected if cytosolic ZAP70-EGFP exchanged during exposure time. These dwell times are slightly longer than estimates of ~10 s obtained from bulk fluorescence recovery after photobleaching (FRAP) experiments (*Bunnell et al., 2002*). However, those experiments also identified a slower-exchanging fraction of ZAP70 that was not included in the average.

ZAP70-EGFP features brighter than single molecules are also observed and we calibrate their stoichiometries using single molecule ZAP70-EGFP intensities from the same cell. For example, the average ZAP70-EGFP feature intensity per frame (136.0 counts ± 0.04 SEM, integrated over a 315 nm square area for each feature) for the cell shown in *Figure 4D* corresponds to 2.9 ± 0.04 SEM ZAP70-EGFP per feature, given that the single molecule intensity is 47 ± 2 counts (SEM; n = 6) in that cell. Each feature therefore contains on average ~6 ZAP70 molecules, since ZAP70-EGFP was expressed in a roughly 1:1 ratio with endogenous ZAP70 (selected by FACS and confirmed by western blotting) in these experiments. Each TCR complex has 10 ITAM domains that, when phosphorylated, can bind one ZAP70 each (*Weiss and Littman, 1994*). Thus observation of ~6 ZAP70 per agonist pMHC suggests only a single TCR is triggered.

Brighter ZAP70-EGFP features tend to be located towards the geometric center of the cell at later time points (>5–10 min after T cell landing). These can be tracked for tens to hundreds of seconds, indicating that recruited ZAP70-EGFP remain stably associated with the pMHC:TCR complex while traveling along the same linear trajectories (*Figure 4A*; *Videos 2 and 3*). The observation of a range of ZAP70-EGFP stoichiometries (from 1 to ~10 ZAP70-EGFP per pMHC:TCR complex) implies that some time-dependent accumulation of ZAP70 is likely to occur, although we have not definitively observed ZAP70-EGFP accumulation over time within individual intensity traces. Taken together, these observations demonstrate that engagement of TCR with an individual agonist pMHC molecule leads to stable association with the actin cytoskeleton, one-to-one TCR triggering (ITAM phosphorylation), and subsequent ZAP70 recruitment. Moreover, since every pMHC:TCR:ZAP70 complex is individually resolved in these experiments, we demonstrate that a single pMHC:TCR complex can lead to TCR triggering without molecular-scale association with other MHC molecules.

## Single molecule agonist pMHC:TCR binding kinetics

Since the slow-moving pMHC can be clearly resolved from the fast moving component, the lifetime of molecules in this bound state is directly observable. The minimum detectable lifetime is limited by the

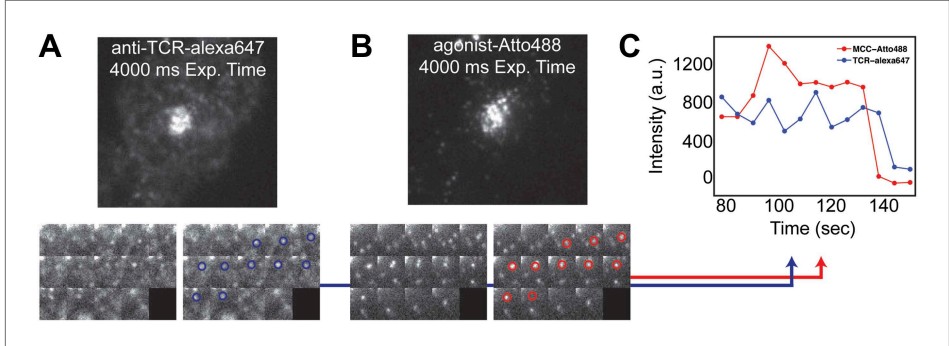

**Figure 2**. TCR and MCC agonist pMHC colocalization. (**A**) and (**B**) TCR and agonist pMHC colocalize in bulk in the central supramolecular assembly cluster (cSMAC) and (**C**) at the single molecule level. Although TCR clusters were not readily observed in these experiments (even using a 4 s camera integration time), previous reports indicate that TCR does cluster at the agonist density (~0.2 molecules/μm²) used here (***Varma et al., 2006***). However, these reports used GPI-MHC, which is problematic because this GPI-linked protein is associated with clustering in supported membranes (***Manz et al., 2011***; ***Dustin and Groves, 2012***). The Ni²⁺-chelating lipids used in supported membranes for the experiments reported here have been shown to increase the likelihood that attached proteins are monodispersed (***Manz et al., 2011***; ***Xu et al., 2011***), and this is confirmed by direct single molecule observation in our experiments.

fastest frame rate (17.5 ms per frame) and the maximum measureable lifetime is limited by photobleaching. For these single molecule tracking experiments, the temporal dynamic range spans from ~50 ms to ~5 min. Unbinding and photobleaching are indistinguishable in fluorescence methods such as this. For molecular binding, characterized by a constant kinetic off-rate, the distribution of observed dwell times, $\tau_{obs}$, is described by

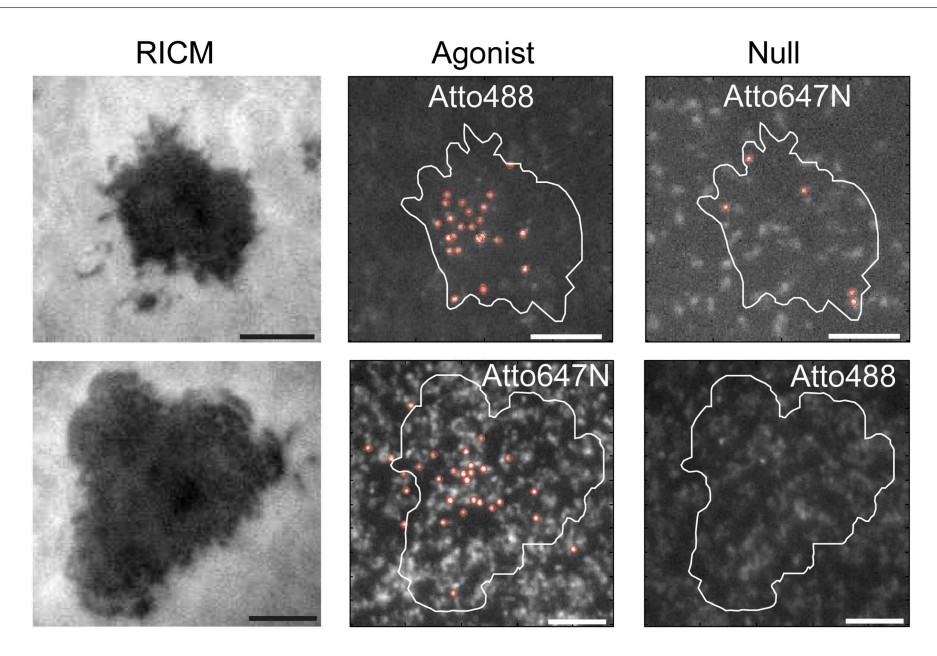

**Figure 3**. Agonist binding is specific and independent of fluorophore. RICM images map the footprint of T cell adhesion to the SLB (mediated through LFA1:ICAM1 binding). T cells engage SLBs conjugated with mixtures of independently labeled MCC agonist and null peptide MHC. Only the MCC agonist pMHC is observed in the slow moving fraction, irrespective of which fluorescent label (Atto488 or Atto647N) it carries.

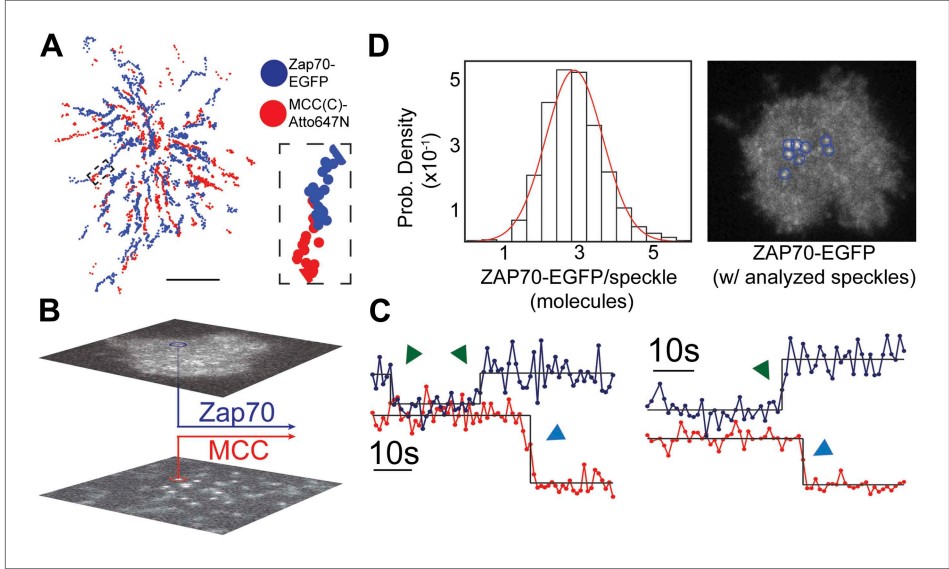

**Figure 4**. ZAP70 recruitment, stoichiometry, and movement are consistent with 1:1 agonist pMHC:TCR stoichiometry. (**A**) A spatial map of MCC-Atto647N single molecule (red) and ZAP70-EGFP (blue) puncta. Raw data are included as **Videos 2 and 3**. Data were recorded at 1 frame/s, such that each adjacent blue or red dot was recorded 1 s apart. Both single MCC agonist pMHC molecules and ZAP70-EGFP puncta follow linear trajectories towards the geometric center of the 2D cell–supported bilayer interface. (**B**) Single ZAP70-EGFP molecules recruited to single agonist pMHC molecules (labeled with MCC-Atto647N) are recorded using sub-pixel color registration (**Figure 4—figure supplement 1**). (**C**) Representative single molecule ZAP70-EGFP (blue) and MCC-Atto647N agonist pMHC (red) localized fluorescence intensity traces. Change-points are detected using a Bayesian change point algorithm (in black; see methods). Step decreases in MCC intensity (red) are most likely agonist pMHC:TCR unbinding events, since $\tau_{off} \ll \tau_{bl}$. Step increases in ZAP70 intensity (blue) are attributed to ZAP70:ITAM binding. (**D**) ZAP70-EGFP puncta brightness histogram is symmetric and centered at 136.0 counts ± 0.04 SEM and corresponds to an average of 2.9 ± 0.04 SEM EGFP molecules (using six single molecule ZAP70-EGFP traces from the same cell for intensity calibration). Bright ZAP70-EGFP speckles are detected from the raw data using an automated algorithm (blue circles; see methods). Scale bar 5 μm.

The following figure supplements are available for figure 4:

**Figure supplement 1**. Dual View color registration.

**Figure supplement 2**. Single molecule ZAP70-EGFP.

$$f\left(\tau_{obs}\right) = \left(\left\langle\tau_{bl}\right\rangle^{-1} + \left\langle\tau_{off}\right\rangle^{-1}\right)e^{-\tau_{obs}\left(\left\langle\tau_{bl}\right\rangle^{-1} + \left\langle\tau_{off}\right\rangle^{-1}\right)},$$

where $\left\langle\tau_{bl}\right\rangle^{-1}$ is the photobleaching rate ($k_b$), $\left\langle\tau_{off}\right\rangle^{-1}$ is the unbinding rate ($k_{off}$), and $\left(\left\langle\tau_{bl}\right\rangle^{-1} + \left\langle\tau_{off}\right\rangle^{-1}\right)^{-1} = \left\langle\tau_{obs}\right\rangle$ is the observed mean dwell time in this experiment. The observed dwell time distributions are roughly exponential, as is expected for molecular binding. Thus by measuring both $k_{bl}$ and $\left\langle\tau_{obs}\right\rangle$ it is possible to determine $\left\langle\tau_{off}\right\rangle$ as long as $\left\langle\tau_{obs}\right\rangle \lesssim \left\langle\tau_{bl}\right\rangle$. We determine $\left\langle\tau_{off}\right\rangle$ to be 53.8 ± 12.2 s for AND and 5.2 ± 0.2 s for 5c.c7 TCRs for Atto488-labeled peptide agonist with $\left\langle\tau_{bl}\right\rangle$ of 300 and 30 s respectively (**Figure 5A,B**). While fluorescent labels can affect binding kinetics, we measure similar values of $\left\langle\tau_{off}\right\rangle$ with both Atto647N and Atto488 labeled peptides (see, e.g., **Figures 1C, 3, and 4A,C**). $\left\langle\tau_{off}\right\rangle$ is also relatively unchanged at high agonist pMHC density (100 molecules/μm²), which is far above minimal levels required for T cell activation and observation of stable TCR microclusters (**Manz et al., 2011**) (**Figure 5C**). We observe that cytoskeleton disruption by the actin-binding molecule, Latrunculin A, moderately increases $\left\langle\tau_{off}\right\rangle$ with the AND TCR and had no significant effect on 5c.c7 kinetics (**Figure 5C**). Similarly, the dwell time distribution was only modestly affected by anti-CD4 (data not shown); however, the total number of TCR:pMHC complexes per cell was smaller in the anti-CD4 experiments, suggesting that the antibody interfered with pMHC:TCR binding.

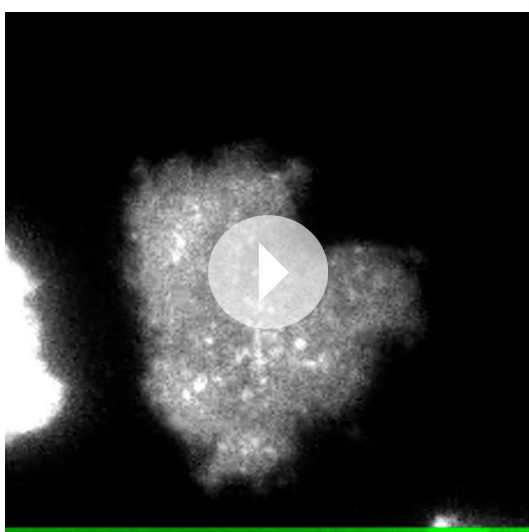

**Video 2**. Simultaneous observation of ZAP70-EGFP recruitment and pMHC:TCR binding immediately after a living AND T cell lands on the SLB. ZAP70-EGFP membrane recruitment (*Video 2*) and pMHC:TCR binding (*Video 3*) occur almost immediately after landing. Radial transport of pMHC:TCR:ZAP70 complexes commences immediately after landing. Data were recorded at 1 frame per second with a 500 ms integration time. These data were analyzed to create the spatial map of pMHC and ZAP70 positions displayed in *Figure 4A*.

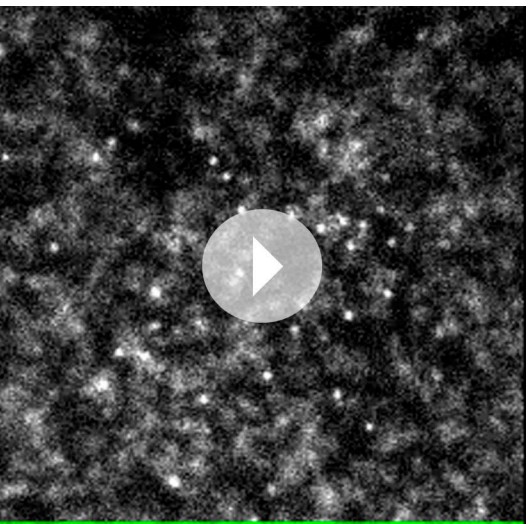

**Video 3**. Simultaneous observation of ZAP70-EGFP recruitment and pMHC:TCR binding immediately after a living AND T cell lands on the SLB. This video shows binding of pMHC:TCR, and is from the same cell as the ZAP70-EGFP data in *Video 2*. Agonist pMHC is labeled as MCC-Atto647N. These data were analyzed to create the spatial map of pMHC and ZAP70 positions displayed in *Figure 4A*.

Tracking observations reveal the time intervals over which individual agonist pMHC molecules remain physically trapped within the immediate vicinity of the same TCR. Thus, although our measured dwell times in live cells are in general agreement with bulk solution measurements of pMHC:TCR kinetic off-rates for both TCRs (*Figure 5C*) (*Corse et al., 2010*; *Newell et al., 2011*), this may not result from the same reasons in each case. Unbinding of pMHC from TCR followed by rapid rebinding to the same TCR or another TCR within the same signaling cluster could conceivably lead to entrapment of pMHC for timescales longer than the lifetime of the molecular interaction. This has been hypothesized as a potential mode by which a small number of agonist pMHC could trigger a larger number of TCR (*Govern et al., 2010*; *Huppa et al., 2010*). Furthermore, recently reported single molecule measurements of pMHC:TCR binding kinetics in live cells, by force probe and by FRET, have suggested accelerated kinetic-off rates (*Huang et al., 2010*; *Huppa et al., 2010*). We investigate this further below.

## Stochastic reaction-diffusion simulations

We quantitatively assess the possibility of serial rebinding of agonist pMHC to multiple TCR within a TCR cluster using a stochastic reaction-diffusion simulation over a large range of $\langle \tau_{off} \rangle$ and TCR cluster size. The total time to escape for an individual molecule, which is the parameter directly measured in pMHC tracking experiments, is given by:

$$\tau_{esc} = \sum_{i=0}^{n} \tau_{off}^{i} + \sum_{i=1}^{n} \tau_{on}^{i} + \tau_{exit}$$

In this representation, $\tau_{off}^{i}$ and $\tau_{on}^{i}$ are the individual dwell times in the bound and unbound configurations, $n$ is the number of rebinding events, and $\tau_{exit}$ is the duration of the final unbound period prior to ultimate escape. For the stochastic simulation, $\tau_{off}^{i}$ and $\tau_{on}^{i}$ are treated as random variables with exponential distributions defined by the in situ measured values of $k_{off}$ and $k_{on}$ for pMHC:TCR binding, respectively. If the pMHC diffuses out of the TCR cluster prior to rebinding, it has escaped. Otherwise, the pMHC rebinds and the cycle repeats. Using the fastest $k_{on}$ (0.17 $\mu m^2 s^{-1}$molecule$^{-1}$) observed in similar hybrid live cell-SLB systems (*Huppa et al., 2010*) and the measured diffusion coefficient of pMHC in our supported membranes, we find that $\tau_{esc} \approx \tau_{off}$ for TCR clusters of the sizes observed experimentally (*Varma et al., 2006*) (≤100 TCR molecules) (*Figure 6*). Only for unrealistically large TCR

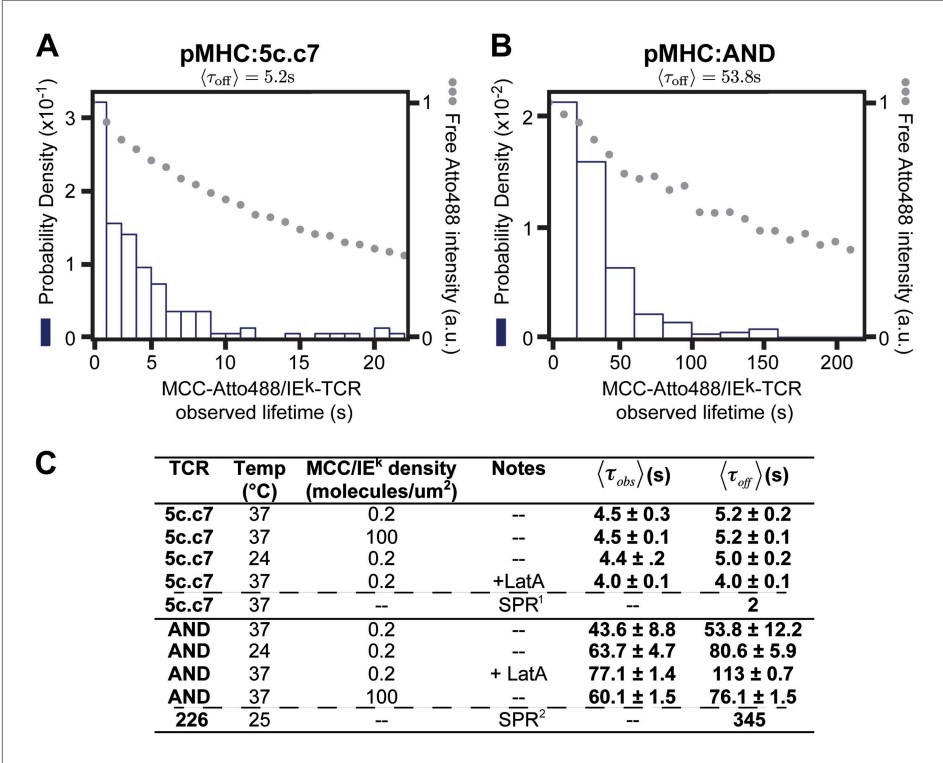

**Figure 5**. The distribution of live cell single molecule agonist pMHC:TCR molecular binding dwell times is observed directly. Measured dwell time distributions for both the 5c.c7 (**A**) and AND (**B**) TCRs are roughly exponential and match reported solution measurements. Bleaching times, $k_{bl}^{-1}$, (grey circles) are measured using agonist pMHC SLB standards without cells and with the same fluorescent label (Atto488) and are significantly longer than observed dwell times, $\tau_{obs}$, for both TCRs. (**C**) Measured values for 5c.c7 and AND CD4+ T cells under varying conditions. Values in columns five and six represent ~300–600 MCC agonist pMHC molecules per experimental condition from a population of 7–20 cells. Data are representative of at least 5 independent experiments performed on T cell blasts isolated from different mice for both the 5c.c7 and AND TCRs. Uncertainty in the average across different mice, shown in columns five and six, is calculated as the standard error of the mean of the molecular averages from different mice. In some cases (e.g. for cytoskeleton disruption experiments with Latrunculin A) one experiment (representative of 7–10 cells and 100 s of single molecule measurements) may be performed, but these data are always compared to a control sample recorded on the same day with T cell blasts from the same mouse. In these cases uncertainty is reported as the standard error of the mean of the molecular dwell time distribution. SPR measurements for 5c.c7 [1](***Huppa et al., 2010***) and AND-related 226 TCRs [2](***Newell et al., 2011***), along with single molecule FRET measurements for 5c.c7 [1](***Huppa et al., 2010***), are shown for comparison.

clusters (~1000 TCR molecules) could rebinding within the same cluster lead to appreciable entrapment $(\tau_{esc} > \tau_{exit})$ (***Figure 6—figure supplement 1***).

These simulations indicate that the observed values of $\tau_{off}$ are unlikely to be the result of rapid serial rebinding of one pMHC with many TCR within a TCR cluster. We note that although TCR clusters are not readily visible at the low antigen densities in these experiments, low level TCR clustering has been reported in the resting state by other methods (***Schamel and Alarcon, 2013***). Even with a very fast $k_{on}$, if pMHC completely disengages from TCR for long enough to diffuse to an adjacent TCR, then the probability of complete escape from the TCR cluster is high. Only in the extreme limit, where $k_{on}$ is so fast that pMHC unbinding is predominantly followed by rebinding to the same TCR, is the escape time appreciably longer than the individual molecular dwell times. The distinction between rapidly rebinding the same TCR and a single engagement is largely semantic. However, it may reveal something about the mechanical stability of pMHC: TCR interactions, and could account for the apparently accelerated $k_{off}$ observed in other types of experiments.

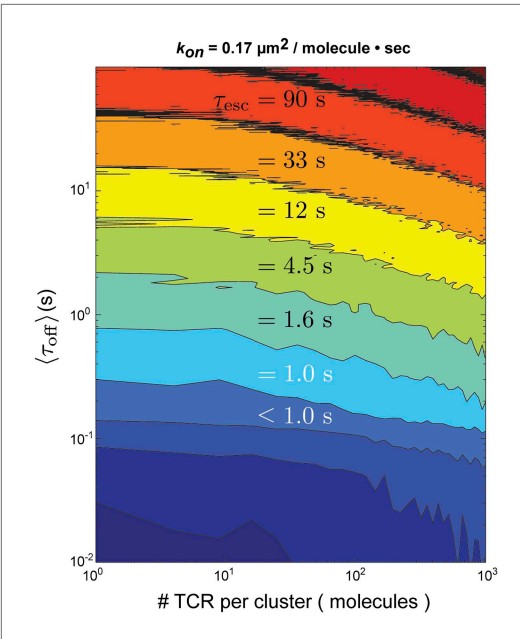

**Figure 6**. Stochastic reaction-diffusion simulation of time before MCC agonist pMHC escape from TCR clusters, $\tau_{esc}$ (in color; log scale), as a function of $\tau_{off}$ and TCR cluster size. For small TCR clusters (1–100 TCR molecules) $\tau_{off} \approx \tau_{esc}$, indicating no serial rebinding. Only for unrealistically large TCR clusters (1000–10,000 molecules) does $\tau_{esc}$ become appreciably longer than $\tau_{off}$.

The following figure supplements are available for figure 6:

**Figure supplement 1**. Stochastic reaction-diffusion simulation of time before MCC agonist pMHC escape from TCR clusters.

## Discussion

All evidence reported here suggests that individual agonist pMHC remain bound to the same TCR for at least several seconds for the 5c.c7 TCR and for approximately one minute for the AND TCR in live cells. Subsequent recruitment of ZAP70 indicates TCR are triggered and movement of the complexes along linear trajectories confirms stable association with the actin cytoskeleton. All of this occurs as a result of a lone agonist pMHC binding TCR, without involvement of other MHC. The measured average pMHC:TCR:ZAP70 stoichiometry indicates TCR triggering is most likely 1:1 with agonist pMHC. Supernumerary triggering of multiple TCR by a single pMHC is not observed on the timescales of minutes investigated in these experiments. The original serial triggering model (**Valitutti et al., 1995**) drew its conclusions from the extent of TCR down regulation measured 5 hr after exposure of T cells to antigen-pulsed APCs. Additionally, the lack of molecular-scale cooperativity in TCR triggering by agonist pMHC indicates that observed cooperativity at the level of cellular calcium response, at similar ligand densities and timescales (**Manz et al., 2011**), is most likely due to intracellular feedback mechanisms (**Altan-Bonnet and Germain, 2005**; **Chan et al., 2004**; **Das et al., 2009**; **Stefanová et al., 2003**).

The timescales of the pMHC:TCR interactions we observe in live cells are consistent with SPR measurements of $k_{off}$ for the 5c.c7 TCR and 226 TCR, which is nearly identical to AND (**Figure 5C** and **Figure 7**) (**Corse et al., 2010**; **Newell et al., 2011**). They are also consistent with reports of a ~2 s kinetic threshold for thymic selection determined in vivo (**Williams et al., 1999**; **Palmer and Naeher, 2009**). Stochastic reaction-diffusion analysis of the measured kinetic and mobility parameters indicates that rapid serial rebinding of agonist pMHC to multiple TCRs in a signaling cluster is unlikely to be a universal mechanism for ligand discrimination and rapid signal amplification in T cells.

The intercellular geometry as well as active processes within the T cell have long been suspected to influence pMHC:TCR interactions (**Shaw and Dustin, 1997**; **Qi et al., 2001**; **Burroughs and Wulfing, 2002**; **Zhu et al., 2013**). Direct in situ measurements of individual pMHC:TCR binding kinetics, such as we report here, are extremely limited (**Huang et al., 2010**; **Huppa et al., 2010**; **Axmann et al., 2012**) but informative comparisons can be made. Notably, a recent intramolecular FRET study of the 5c.c7 TCR binding MCC pMHC reports short ($\langle \tau_{off} \rangle$~150 ms) in situ 2D dwell times, nearly 35 times faster than the $\langle \tau_{off} \rangle$ = 5.2 ± 0.2 s we measured by tracking. This same study measures $\langle \tau_{off} \rangle$ 25 times longer under conditions of cytoskeleton disruption, essentially in agreement with tracking observations. This result has been interpreted to mean that the actin cytoskeleton actively destabilizes agonist pMHC:TCR complexes. In contrast, we observe very small effects of actin disruption on agonist pMHC dwell times for AND and essentially no effect with 5c.c7 TCR (**Figure 5C**).

It is conceivable that the agonist pMHC:TCR complex does not remain bound in the same structural configuration for the duration of engagement. If mechanical coupling to actin significantly reduces the apparent $\langle \tau_{off} \rangle$ in a single molecule FRET measurement, but not in a single molecule tracking experiment, this raises the possibility that mechanical forces can induce conformational alterations in agonist pMHC:TCR without complete disengagement of the complex. Recent structural studies of pMHC:TCR indicate the possibility of such flexibility (**Adams et al., 2011**; **Hawse et al., 2012**; **Reboul et al., 2012**).

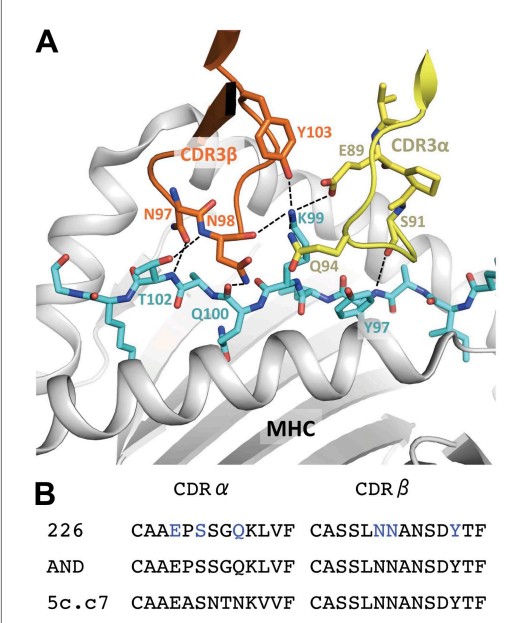

**Figure 7**. Interaction of the 226 TCR with the MCC peptide. (**A**) CDR3α (yellow) and CDR3β (orange) loops form all the specific interactions with the peptide (cyan). Hydrogen bonds between the CDR3 loops and the MCC residues are shown by dashed lines (PDB, 3QIU) (**B**) Comparison of CDR3 loops between the 226 and AND TCRs reveal identical sequences and suggest similar binding kinetics. 226 and AND also share Vα, Jα, Vβ, and Jβ gene segments that encode the residues specific for interaction with IE[k] (*Newell et al., 2011*). The residues involved in hydrogen bonds between 226 and MCC are shown in blue.

While putative conformational changes would not affect single-molecule tracking measurements, they could produce a FRET signature (*Ha et al., 1999*; *Majumdar et al., 2007*). Under such a scenario, apparently fast $\langle \tau_{off} \rangle$ observed by FRET imaging (*Huppa et al., 2010*) may not correspond to actual molecular unbinding and escape of agonist pMHC from TCR.

In the aggregate, the data reported here indicate that spatially discrete pMHC:TCR:ZAP70 complexes form according to molecular mass action laws with relatively predictable chemical kinetics and stoichiometry in living cells. The observed pMHC:TCR molecular binding kinetics mirror solution measurements and we observe no evidence for molecular scale cooperativity in the triggering of TCR by agonist pMHC (at low agonist density). Thus any amplification or other form of signal processing must occur downstream of TCR triggering.

## Materials and methods

### DNA, protein, and T cell preparation

A plasmid containing enhanced green fluorescent protein fused to CD3 zeta-chain-associated protein of 70 kDa (*Zap70-EGFP*) was a gift of Takashi Saito, RIKEN Research Center for Allergy and Immunology, Yokohama, Japan (*Yokosuka et al., 2005*). The *Zap70-EGFP* gene was amplified by PCR and subcloned into a murine stem cell virus parent vector (pMSCV).

Bi-hexahistidine-tagged major histocompatibility complex (MHC) class II I-E[k] protein was produced and purified as previously described

(*Nye and Groves, 2008*). A decahistidine-tagged *ICAM1-TagBFP* fusion protein was generated by PCR amplifying the TagBFP sequence (Evrogen Inc., Moscow, Russia) and subcloning it into a pN1-ICAM1 vector. The entire *Icam1-TagBFP* gene was then further subcloned into the pFastBac1 vector (Invitrogen Inc., Carlsbad, CA), which was used to generate baculovirus for infection of High Five cells (Invitrogen Inc.). The ICAM1-TagBFP was subsequently purified on a Ni-NTA-agarose affinity column, eluted with an imidazole gradient, dialyzed, and stored in Tris buffer containing 10% glycerol. AND CD4+ T cells (*Kaye et al., 1989*) and 5 c.c7 CD4+ T cells were harvested and cultured essentially as previously described (*Smith et al., 2011*; *Smoligovets et al., 2012*).

T cells were transduced with *Zap70-EGFP* and sorted using fluorescence-activated cell sorting (FACS) according to viability and EGFP expression. The population of transduced cells that was used expressed EGFP at no more than 50% of the highest EGFP level in the overall EGFP-positive population.

### Peptide purification and labeling

Using the basic sequence of moth cytochrome c (amino acids 88–103) and previously described variants (*Huppa et al., 2010*), the following peptides were synthesized by David King at the HHMI Mass Spectrometry Laboratory at UC Berkeley and/or commercially (Elim Biopharmaceuticals, Hayward, CA): MCC (ANERADLIAYLKQATK), MCC(C) (ANERADLIAYLKQATKGGSC), MCC-null(C) (ANERAELIAYLTQAAKGGSC). For fluorophore labeling, cysteine-containing peptides were dissolved in a small amount of phosphate buffer and mixed in a 1:2 molar ratio with Atto 647N resuspended in a small amount of 1-propanol or lyophilized Atto 488 (Atto-Tec GmbH, Siegen, Germany) and labeled using maleimide-thiol chemistry. The peptides were then incubated at room temperature for at least

1 hr and purified on a C18 reverse phase column (Grace–Vydac, Deerfield, IL) and H$_2$O:acetonitrile gradient using ÄKTA explorer 100 FPLC system (Amersham Pharmacia Biotech, Piscataway, NJ). Peptide identity was confirmed after purification using mass spectrometry.

## Microscopy

TIRF experiments were performed on an inverted microscope (Nikon Eclipse Ti; Technical Instruments, Burlingame, CA) using a custom-built laser launch with 488 nm (Sapphire HP; Coherent Inc., Santa Clara, CA) and 640 nm (Cube; Coherent Inc.) diode lasers, as described previously (*Smith et al., 2011*). Laser powers measured at the sample were 0.8 mW (640 nm) and 0.5 mW (488 nm) for 500 ms exposures and 4.4 mW (640 nm) for 17.5 ms exposures. A dichroic beamsplitter (z488/647rpc; Chroma Technology Corp., Bellows Falls, VT) reflected the laser light through the objective lens (Nikon 1.49 NA TIRF; Technical Instruments, Burlingame, CA) and fluorescence images were recorded using an EM-CCD (iXon 597DU; Andor Inc., South Windsor, CT) after passing through a laser-blocking filter (Z488/647 M; Chroma Technology Corp., Bellows Falls, VT). Bandpass filters (FF03 525/50; Semrock Inc., Rochester, NY and ET 700/75, Chroma Technology Corp., Bellows Falls, VT) were placed in a DualView 2 Simultaneous Imaging System (Photometrics, Tuscon, AZ). Colors were registered before every two-color experiment by imaging 100 nm Tetraspec beads (Invitrogen Inc.) deposited on a coverslip patterned with a Cr grid with ~80 nm width and 3–4 µm pitch Cr lines, since the Tetraspec beads preferentially bind the regular Cr pattern. Exposure times and time-lapse periods for most experiments were set using image collection software (MetaMorph 7.5; Molecular Devices Inc., Downingtown, PA), which drives an external shutter (Uniblitz LS6; Vincent Associates, Rochester, NY). Exposure time and Fast Kinetics Mode for short (17.5 ms) integration time experiments were set using Andor Solis (Andor Inc., South Windsor, CT). Exposure times were measured directly from the Fire output of the EM-CCD using an oscilloscope (TDS 210; Tektronix, Inc., Beaverton, OR).

## Imaging chamber and supported lipid bilayer preparation

Single unilamellar vesicles (SUVs) were formed by tip sonication of a solution composed of 98 mol % 1,2-dioleoyl-sn-glycero-3-phosphocholine (DOPC) and 2 mol % 1,2-dioleoyl-sn-glycero-3-[(N-(5-amino-1-carboxypentyl) iminodiacetic acid) succinyl] (nickel salt) (Ni$^{2+}$-NTA-DOGS) (Avanti Polar Lipids, Alabaster, AL) in Mill-Q water (EMD Millipore, Billerica, MA). Tip sonication was preferred to vesicle extrusion due to the introduction of significant levels of fluorescent impurities into the SUVs during extrusion. Prior to experiments, #2 40 mm diameter round coverslips were ultrasonicated for 30 min in 50:50 isopropyl alcohol:water, rinsed thoroughly in Milli-Q water (EMD Millipore, Billerica, MA), etched for 5 min in piranha solution (3:1 sulfuric acid:hydrogen peroxide), and again rinsed thoroughly in Milli-Q water. The coverslips were used in the assembly of FCS2 Closed Chamber Systems (flow cells; Bioptechs, Butler, PA), which were pre-filled with Tris-buffered saline (TBS; 19.98 mM Tris, 136 mM NaCl, pH 7.4; Mediatech Inc., Herndon, VA). SUVs were then flowed into the chambers, and bilayers were allowed to form for at least 30 min. The bilayers were rinsed once with TBS, incubated for 5 min with 100 mM NiCl$_2$ in TBS, rinsed with TBS, and then rinsed with a T cell imaging buffer composed of 1 mM CaCl$_2$, 2 mM MgCl$_2$, 20 mM HEPES, 137 mM NaCl, 5 mM KCl, 0.7 mM Na$_2$HPO$_4$, 6 mM d-glucose, and 1% wt/vol bovine serum albumin. 48 hr prior to experiments, MHC was loaded with peptide at 37°C in a buffer composed of 1% wt/vol bovine serum albumin in phosphate-buffered saline and brought to pH 4.5 with citric acid. Unbound peptide was separated from peptide loaded MHC (pMHC) using 10k spin concentrators (Amicon Ultra, Cork, Ireland) and then pMHC was diluted in imaging buffer. ICAM1-TagBFP and pMHC were further diluted with imaging buffer, introduced into the flow cells, and incubated for 35 min followed by a rinse with imaging buffer. T cells resuspended in imaging buffer and added to the flow cells 35 min after the final rinse and imaged immediately for 30–60 min. To visualize TCR, T cells were incubated in a solution of 1 µl Alexa 647 (Invitrogen Inc.)-labeled H57 anti-TCR Fab and 100 µl imaging buffer for 20 min at 4°C prior to the regular imaging buffer resuspension. All other incubations during this protocol were performed at room temperature, and imaging experiments were performed at 37°C.

## Data analysis

Single molecule diffraction-limited spots were detected in raw .tif image stacks of agonist pMHC labeled with MCC-Atto488 and MCC-647N molecules by filtering for both size and intensity and linked into tracks using published particle detection and tracking algorithms (*Crocker and Grier, 1996*) adapted for MATLAB (The Mathworks; Natick, MA) by Daniel Blair and Eric Dufresne

(http://physics.georgetown.edu/matlab/; accessed 16 August 2012). Size and intensity thresholds were first determined by eye using a test data set and then applied uniformly to all data collected with the same exposure time and incident laser intensities. Single molecules were identified by step photobleaching detected in an automated way using a Bayesian change point detection algorithm (*Ensign and Pande, 2009*).

The brightness of ZAP70-EGFP features varies from a single molecule to several molecules, and different brightness features are detected using slightly different methods, despite the fact that the features are physically similar. Bright ZAP70-EGFP features (as shown in *Figure 4A,D*) were detected using the same algorithm as is used for single molecule pMHC. The lower signal-to-noise single molecule ZAP70-EGFP intensity traces like those in *Figure 4C* were obtained by summing the intensity of the ZAP70-EGFP channel using the agonist pMHC (labeled with Atto647N) molecule position as a mask, as is explained in the main text.

The lifetime of the bright ZAP70-EGFP speckles is difficult to accurately assess due to the fluctuating background and varying speckle intensity (which biases measurement towards brighter, longer-lived fluorescent features), but speckle lifetimes appear to be longer than the single molecule ZAP70-EGFP lifetimes. Single molecule ZAP70-EGFP molecules are uncorrected for photobleaching of both ZAP70-EGFP and agonist pMHC and therefore the range of binding times reported (12–107 s) only serves as a lower bound for the molecular ZAP70 dwell time.

The agonist pMHC step size distribution at 17.5 ms resolution in *Figure 1D* is populated using similar particle detection and tracking methods to the 500 ms resolution analysis., Agonist pMHC:TCR binding kinetics cannot be uniquely inferred from the step size distribution, since the step size distribution is a time-independent analysis. For instance, the step size distribution measured over a certain time period with $2k_{on}$ and $2k_{off}$ would appear identical to a scenario with $k_{on}$ and $k_{off}$.

Lifetime distributions are roughly exponential and of the form $f\left(\tau_{obs}\right) = \left\langle\tau_{obs}\right\rangle^{-1} e^{-\tau_{obs}/\left\langle\tau_{obs}\right\rangle}$, where $\tau_{obs}$ is the observed dwell time in our experiments. The individual kinetic transitions were derived assuming the following model:

$$pMHC_{free} \underset{k_{off}}{\overset{k_{on}}{\rightleftharpoons}} pMHC:TCR \xrightarrow{k_{bl}} pMHC_{bleached}$$

where $pMHC_{free}$ is the fast-mobility state, $pMHC:TCR$ is the slow-mobility state (or the TCR stably bound state), $pMHC_{bleached}$ is the bleached slow-mobility state, $k_{off}$ and $k_{on}$ are the rates of transitions between the bound and the free pMHC, and $k_{bl}$ is the rate of transition from bound pMHC:TCR to photobleached pMHC. We assume that transitions between states follow a Markov memory-less process and derive a probability density function, $f\left(\tau_{obs}\right)$, for the single molecule dwell time distribution: $f\left(\tau_{obs}\right) = \left(\left\langle\tau_{bl}\right\rangle^{-1} + \left\langle\tau_{off}\right\rangle^{-1}\right) e^{-\tau_{obs}\left(\left\langle\tau_{bl}\right\rangle^{-1} + \left\langle\tau_{off}\right\rangle^{-1}\right)}$, where $\left(\left\langle\tau_{bl}\right\rangle^{-1} + \left\langle\tau_{off}\right\rangle^{-1}\right)^{-1} = \left\langle\tau_{obs}\right\rangle$ is the observed mean dwell time in our experiments. Agonist pMHC labeled with Atto488 and Atto647N SLB bleaching curves were background subtracted and then fit to an exponential decay function of the form $f\left(t\right) = k_{bl}e^{-k_{bl}t}$. Fitting was done using MATLAB.

## Stochastic kinetic simulation

Simulations were performed using MATLAB. Our simulation models a TCR cluster as a square lattice upon which agonist pMHC molecules bind discrete TCR lattice sites for duration $\tau_{off}^i$, $\tau_{off}^i$ where is treated as a random variable drawn from an exponential distribution with mean equal to $\left\langle\tau_{off}\right\rangle$. $\left\langle\tau_{off}\right\rangle$ is varied over several orders of magnitude and is chosen to match measured values from the literature. pMHC are initially placed at a randomly lattice position drawn from a uniform distribution. After each time period (determined by $\tau_{off}^i$), the agonist pMHC molecule steps to a new lattice site or stays at the same lattice site (the lattice spacing is set to 10 nm to roughly follow the size of the TCR complex [*Newell et al., 2011*; *Yin et al., 2012*]) until the agonist pMHC is no longer on the TCR cluster, such that $\tau_{esc}^k = \sum_{i=0}^{n} \tau_{off}^i + \sum_{i=1}^{n} \tau_{on}^i + \tau_{exit}$, where $n$ indicates the number of steps an individual agonist pMHC molecule takes before exiting the TCR cluster, $\tau_{on}$ is the time period between unbinding and binding events, and $\tau_{exit}$ is the time between the last unbinding event and the ultimate exit from the TCR cluster. Step size is treated as a combination of two independent random variables, $(\Delta x, \Delta y,)$ drawn from Gaussian distributions with mean 0 and standard deviation $\sqrt{2D_{SLB}\tau_{on}}$. The step size is then a random variable $\Delta r = \sqrt{\Delta x^2 + \Delta y^2}$ and the angle of displacement is drawn from a uniform distribution.

The interval between binding events, $\tau_{on}$, is treated as a random variable drawn from an exponential distribution with mean $k_{on}\,\rho_{TCR}$ where the density of TCR, $\rho_{TCR}$, is taken to be 10,000 molecules/μm² (as in the central supramolecular activation cluster). In this way $\langle \tau_{off} \rangle = \frac{1}{k}\sum_k \tau_{esc}^k$, where $k$ is the number of iterations (100 in the case of *Figure 6* and *Figure 6—figure supplement 1*), is calculated for every combination of $\langle \tau_{off} \rangle$, TCR cluster size, and $k_{on}$. Note that since $\tau_{off} \gg \tau_{on}$, $\frac{\tau_{esc}}{\tau_{off}} \approx \langle n \rangle$, where $\langle n \rangle$ is equivalent to the TCR cluster size. This relationship between $\frac{\tau_{esc}}{\tau_{off}}$ and TCR cluster size can be seen in *Figure 6—figure supplement 1*. It is possible that agonist pMHC binding interactions with CD4 could slow the mobility of an individual agonist pMHC within a TCR cluster relative to $D_{SLB}$ when the agonist pMHC are unbound from TCR. This could hypothetically lead to entrapment and long single molecule tracks (like those reported here) in the absence of direct, sustained agonist pMHC-TCR interactions. While such a mechanism is conceivable, there is no direct evidence for such a tethering mechanism in the literature.

## Acknowledgements

The content is solely the responsibility of the authors and does not necessarily represent the official views of the National Institutes of Health. We thank David King of the HHMI Mass Spec Facility for peptide synthesis and Mass Spectroscopy, Nicole Fay, Kate Alfieri, and Niña Hartman for MHC purification, and Brian D Belardi for a critical reading of the manuscript.

## Additional information

### Funding

| Funder | Grant reference number | Author |
| --- | --- | --- |
| National Institute of Allergy and Infectious Diseases | PO1 AI091580 | Geoff P O'Donoghue, Rafal M Pielak, Alexander A Smoligovets, Jenny J Lin, Jay T Groves |

The funders had no role in study design, data collection and interpretation, or the decision to submit the work for publication.

### Author contributions

GPO, RMP, Conception and design, Acquisition of data, Analysis and interpretation of data, Drafting or revising the article; AAS, JJL, Acquisition of data, Drafting or revising the article; JTG, Conception and design, Analysis and interpretation of data, Drafting or revising the article

### Ethics

Animal experimentation: CD4+ T-cell blasts were cultured from the lymph nodes and spleens of first generation AND x B10.BR and 5c.c7 TCR transgenic mice on day 1 of the T-cell protocol in accordance with Lawrence Berkeley National Laboratory Animal Welfare and Research Committee-approved protocol 17702.

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
