## [Decision Letter]

Thank you for sending your work entitled “Direct single molecule measurement of TCR triggering by agonist pMHC in living primary T cells” for consideration at *eLife*. Your article has been favorably evaluated by a Senior editor, Detlef Weigel, and 3 reviewers, one of whom is a member of our Board of Reviewing Editors.

The Reviewing editor and the other reviewers discussed their comments before we reached this decision, and the Reviewing editor has assembled the following comments to help you prepare a revised submission.

Overall, this is an outstanding study, one that has made a number of technical breakthroughs and emerges with interesting results. The authors examine the interactions between pMHC and TCR, at a single molecule level, which provides a direct measurement on the association times of this complex. They also examine how the pMHC-TCR complex recruits ZAP70, which is a direct measure of triggering and the initiation of signal transduction. This is a very nicely executed two color, single molecule experiment. There are several interesting findings: 1) the TCR-pMHC act as discrete entities and do not appear to aggregate; 2) the association times differ 10-fold for two different peptide agonists, revealing distinct information encoded by off-rates; 3) the dissociation rates are much slower than was previously reported by a live cell FRET measurement (the new data are more direct and believable, and the information will be important for the field); 4) Virtually all pMHC-TCR complexes become immobilized and hook onto the actin cytoskeleton; and 5) each pMHC-TCR recruits several ZAP70 molecules and these complexes also are extremely long lived.

Major comments:

1) This manuscript follows others (18; 17) that have tried to address this problem and come away with interesting insights. The current study takes a unique approach using low intensity light for longer time periods, which selects against short-term interactions to track TCR-PMHC interactions and comes up with some contradictions with the published work. These differences to the previously published results are puzzling and need to be discussed.

2) There is not much of a difference between 24 and 37 degrees. This is surprising and should be discussed.

3) The Introduction could be improved for a general audience. Currently it reads like a longer version of the Abstract (or a shorter version of the Results). This section would be a good place to set up the main issues of the paper, e.g., why off-rates are important and what attempts have been made to measure this in the past. Also a few small points – are all pMHC-TCR-Zap70 complexes attached to actin or is there evidence of them being formed, diffusion, and then joining onto the actin treadmill? Can one detect variability or successive recruitment of Zap70? It is stated that there is an average of 6 but we are wondering if one could extract additional information at the single molecule level of clear variability (particularly for two different pMHC) or accumulation of Zap70 over time.

---

## [Author Response]

*1) This manuscript follows others (18; 17) that have tried to address this problem and come away with interesting insights. The current study takes a unique approach using low intensity light for longer time periods, which selects against short-term interactions to track TCR-PMHC interactions and comes up with some contradictions with the published work. These differences to the previously published results are puzzling and need to be discussed*.

The differing results with respect to in situ pMHC:TCR kinetics observed by tracking in our experiments and by smFRET in the [18] paper are indeed puzzling. Several key points regarding this issue are contained in the original manuscript. We highlight these below and have made additional clarifications in the text (e.g., paragraph 1 in “Single molecule agonist pMHC:TCR binding kinetics”).

The variable exposure times used in our experiments allow observations to be directed at different time scales. Thus although much of the data presented was acquired under conditions that do indeed select against short-term interactions, we have also done the corresponding high-speed measurements (see, for example, the single molecule trace in Figure 1). The fast pMHC:TCR kinetics reported in Huppa et al., are still an order of magnitude slower than our fastest time resolution (17.5 ms); such fast events are not likely to be missed in these tracking experiments using the fast exposure times.

We devote several analyses in the manuscript towards possible differences in what is measured by tracking vs. smFRET. Specifically, we consider (and rule out) serial rebinding of a single pMHC to multiple TCR in a TCR cluster (“Stochastic reaction-diffusion simulations”). In the Discussion (paragraphs 3 and 4) we suggest structural flexibility within the pMHC:TCR complex could explain the differing observations, particularly since the fast kinetics are only observed when the pMHC:TCR complexes are under strain. This could easily obliterate a smFRET signal even without the complex fully disengaging (25). Such flexibility is supported by recent structural analyses of pMHC:TCR (1; 16; 34). Therefore, with the information at hand, we suggest this is the most likely explanation. It is important to note that our measurements agree with solution SPR measurements ([7] & ; [29]) and with in vivo requirements for negative selection (44; 31) for two different TCR. We do not analyze measurements made with a different pMHC:TCR combination (17) in detail because they are not directly comparable.

*2) There is not much of a difference between 24 and 37 degrees. This is surprising and should be discussed*.

There is only a modest difference between dwell times measured at 24 and 37°C. The difference is more pronounced for the MCC-AND interaction, 54 s vs 81 s at 24 and 37°C, respectively. We observe a slight temperature effect in the 5c.c7 system; however, the intrinsic experimental error in comparing the dwell time distributions at 24 and 37°C is higher for shorter interactions. The effect is also easier to see for the MCC-AND interaction because there are more MCC-AND complexes per T cell (data not shown). It is also possible that changes in cellular behavior in response to temperature could also affect these measurements in ways that are not observed in solution measurements with purified components. At 24°C T cells have a smaller surface area on the supported membrane than at 37°C, are slower to land, and also have decreased lamellipodial motion. These factors could affect the environment of the pMHC:TCR interaction in unpredictable ways and perhaps could compensate for the direct temperature effect on molecular interactions.

*3) The Introduction could be improved for a general audience. Currently it reads like a longer version of the Abstract (or a shorter version of the Results). This section would be a good place to set up the main issues of the paper, e.g., why off-rates are important and what attempts have been made to measure this in the past*.

We have expanded and clarified the Introduction to better introduce the importance of quantitative single molecule measurements (e.g., of kinetic off-rate, stoichiometry, etc.) to the understanding of TCR signaling mechanisms.

*Also a few small points – are all pMHC-TCR-Zap70 complexes attached to actin or is there evidence of them being formed, diffusion, and then joining onto the actin treadmill? Can one detect variability or successive recruitment of Zap70? It is stated that there is an average of 6 but we are wondering if one could extract additional information at the single molecule level of clear variability (particularly for two different pMHC) or accumulation of Zap70 over time*.

All observed pMHC:TCR:ZAP70 complexes are transported radially towards the geometric cell center (Videos 1, 2 and 3; Figure 3) of the 2-dimensional cell-supported membrane interface. We do not observe evidence for pMHC:TCR:ZAP70 random diffusion prior to radial transport on these timescales – such sequential events could occur on faster timescales.

Clear variability does exist in the number of ZAP70-EGFP per pMHC:TCR complex. This is most clearly shown in Figure 3, where the intensity distribution of ZAP70-EGFP puncta is symmetric and unimodal with an average of 2.9 ZAP70-EGFP/puncta. The distribution of ZAP70-EGFP intensities (from single ZAP70 molecules up to ∼10) implies that time-dependent ZAP70-EGFP accumulation must occur, but we have not definitively observed accumulation of ZAP70-EGFP over time in one intensity trace. The observation of single molecule ZAP70-EGFP accumulation in a single intensity trace (i.e., several consecutive step intensity increases) is technically challenging due to the convolution of several time-dependent processes: MCC-Atto647N bleaching, ZAP70-EGFP bleaching, pMHC:TCR unbinding, ZAP70-ITAM binding, and ZAP70-ITAM unbinding. We do, however, make the bulk observation that the brightest ZAP70-EGFP features are observed at later time points. The text has been clarified accordingly (“TCR triggering monitored by ZAP70 recruitment”).